# Hypomyelinating Leukodystrophy 15 (HLD15)-Associated Mutation of EPRS1 Leads to Its Polymeric Aggregation in Rab7-Positive Vesicle Structures, Inhibiting Oligodendroglial Cell Morphological Differentiation

**DOI:** 10.3390/polym13071074

**Published:** 2021-03-29

**Authors:** Sui Sawaguchi, Mizuki Goto, Yukino Kato, Marina Tanaka, Kenji Tago, Hiroaki Oizumi, Katsuya Ohbuchi, Kazushige Mizoguchi, Yuki Miyamoto, Junji Yamauchi

**Affiliations:** 1Laboratory of Molecular Neurology, Tokyo University of Pharmacy and Life Sciences, Hachioji, Tokyo 192-0392, Japan; s167023@toyaku.ac.jp (S.S.); s179035@toyaku.ac.jp (M.G.); s179026@toyaku.ac.jp (Y.K.); swgchsui@gmail.com (M.T.); 2Department of Biochemistry, Jichi Medical University, Shimotsuke, Tochigi 321-0498, Japan; ktago@jichi.ac.jp; 3Tsumura Research Laboratories, Tsumura & Co., Inashiki, Ibaraki 200-1192, Japan; ooizumi_hiroaki@mail.tsumura.co.jp (H.O.); oobuchi_katsuya@mail.tsumura.co.jp (K.O.); mizoguchi_kazushige@mail.tsumura.co.jp (K.M.); 4Department of Pharmacology, National Research Institute for Child Health and Development, Setagaya, Tokyo 157-8535, Japan; miyamoto-y@ncchd.go.jp

**Keywords:** Pelizaeus–Merzbacher disease, EPRS1, Rab7, oligodendrocyte, differentiation, myelination

## Abstract

Pelizaeus–Merzbacher disease (PMD), also known as hypomyelinating leukodystrophy 1 (HLD1), is an X-linked recessive disease affecting in the central nervous system (CNS). The gene responsible for HLD1 encodes proteolipid protein 1 (plp1), which is the major myelin structural protein produced by oligodendroglial cells (oligodendrocytes). HLD15 is an autosomal recessive disease affecting the glutamyl-prolyl-aminoacyl-tRNA synthetase 1 (*eprs1*) gene, whose product, the EPRS1 protein, is a bifunctional aminoacyl-tRNA synthetase that is localized throughout cell bodies and that catalyzes the aminoacylation of glutamic acid and proline tRNA species. Here, we show that the HLD15-associated nonsense mutation of Arg339-to-Ter (R339X) localizes EPRS1 proteins as polymeric aggregates into Rab7-positive vesicle structures in mouse oligodendroglial FBD-102b cells. Wild-type proteins, in contrast, are distributed throughout the cell bodies. Expression of the R339X mutant proteins, but not the wild-type proteins, in cells induces strong signals regulating Rab7. Whereas cells expressing the wild-type proteins exhibited phenotypes with myelin web-like structures bearing processes following the induction of differentiation, cells expressing the R339X mutant proteins did not. These results indicate that HLD15-associated EPRS1 mutant proteins are localized in Rab7-positive vesicle structures where they modulate Rab7 regulatory signaling, inhibiting cell morphological differentiation. These findings may reveal some of the molecular and cellular pathological mechanisms underlying HLD15.

## 1. Introduction

In the central nervous system (CNS) and the peripheral nervous system (PNS), myelin sheaths derived from differentiated oligodendroglial cell (oligodendrocyte) or Schwann cell plasma membranes wrap neuronal axons in multiple layers of myelin. The surface areas of mature myelin sheaths end up much larger than those of the plasma membranes of mature oligodendrocytes [1,2,3,4]. Mature myelin not only speeds the propagation of saltatory conduction but also protects axons from various physical and physiological stresses [1,2,3,4].

Hypomyelinating leukodystrophies (HLDs) are a recently classified group of hereditary neuropathies that are primarily linked to oligodendrocytes. These diseases are very rare, affecting one out of every 200,000 to 500,000 people. Recent nucleotide sequencing technologies with next-generation sequencing have enabled us to identify expected and unexpected HLD-responsible genes and HLD-related disease-associated genes [see Table 1; Refs. 9 and 10]. Pelizaeus–Merzbacher disease (PMD), also called HLD1, is a prototypic HLD [5,6,7,8]. The gene responsible for HLD1 encodes proteolipid protein 1 (PLP1), the major myelin structural protein produced by oligodendrocytes [5,6,7,8]. The gene responsible for HLD15, in contrast, is the glutamyl-prolyl-aminoacyl-trna synthetase 1 (*eprs1*) gene, which encodes the EPRS1 protein, a bifunctional aminoacyl-tRNA synthetase that catalyzes the aminoacylation of glutamic acid and proline tRNA species. HLD15-associated mutation of EPRS1 proteins results in decreased translation capacity, leading to insufficient myelination in the developing brain [9,10,11]. In brain imaging, HLD15 appears as a hypomyelinating leukodystrophy with thin corpus callosum. HLDs involving HLD15 are diagnosed with reference to both clinical and magnetic resonance imaging (MRI) features followed by genetic confirmation [9,10,11]. The features of HLD15 include significant visual loss with variable amblyopia and dysphagia as well as dystonia, ataxia, and spasticity, although the severity of the disorder is variable. Most HLD15 patients develop severe optic atrophy and some have hearing loss [11]. To date, no therapeutic strategy has been established, and symptomatic treatment, such as anticonvulsant medications for patients with spasticity, is the typical response [5,6,7,8,9,10,11].

While HLD15-associated mutations of EPRS1 decrease EPRS protein expression levels [11], some residual mutant proteins remain, yet it is unclear how these mutant proteins cause the disease’s molecular and cellular pathological effects. Here we show that the HLD15-associated nonsense mutation of Arg339-to-Ter (R339X) localizes EPRS1 proteins as polymeric aggregates into vesicle structures that are positive for Rab7 in the mouse oligodendroglial FBD-102b cells that we have used as model oligodendrocytes. Wild-type EPRS1 proteins, in contrast, are localized throughout cell bodies. In addition, signaling that regulate Rab7 were specifically upregulated specifically by the expression of EPRS1 mutant proteins. Cells expressing the mutant proteins had decreased morphological differentiation: whereas cells expressing the wild-type proteins exhibited a phenotype with myelin web-like structures, cells expressing the mutant proteins did not. These results suggest a possible pathological mechanistic basis of HLD15 at the molecular and cellular levels.

## 2. Material and Methods

### 2.1. Primary and Secondary Antibodies

The following antibodies were used: mouse monoclonal antibody against a KDEL-containing peptide of the endoplasmic reticulum (ER)-resident glucose-regulated protein (GRP78) (Cat. No. M181-3; immunofluorescence [IF], 1/200), rabbit polyclonal myelin basic protein (MBP; Cat. No. PD004; immunoblotting [IB], 1/500), mouse monoclonal anti-actin (the ACTB subtype; Cat. No. M177-3; IB, 1/40,000), mouse monoclonal anti-DDDDK antigen (also called anti-FLAG antigen; Cat. No. M185-3; IB, 1/20,000 and IF, 1/1000), and rabbit polyclonal anti-DDDDK antigen (Cat. No. PM020; IF, 1/1000) from MBL (Aichi, Japan); mouse monoclonal anti-Golgi matrix protein 130 (GM130) (Cat. No. 610822; IF, 1/200) from BD Biosciences (Franklin Lakes, NJ, USA); rabbit polyclonal anti-late endosomal Rab7 (Cat. No. 9367S; IF, 1/100) from Cell Signaling Technology (Danvers, MA, USA); mouse monoclonal anti-lysosomal-associated membrane protein 1 (LAMP1) (Cat. No. ab25630; IF, 1/100) and rabbit monoclonal anti-Rab9 (Cat. No. ab179815; IF, 1/100) from Abcam (Cambridgeshire, UK); and mouse monoclonal anti-Sox10 (Cat. No. sc-365692; IB, 1/500), mouse monoclonal anti-CCZ1 homolog, vacuolar protein trafficking and biogenesis associated (CCZ1; Cat. No. sc-514290; IB, 1/50), and mouse monoclonal anti-TRE2, Bub2, and CDC16 homology domain 1 family member 5 (TBC1D5; Cat. No. sc-376296; IB, 1/50) from Santa Cruz Biotechnology (Santa Cruz, CA, USA).

The following secondary antibodies were used: anti-rabbit or mouse IgG F(ab’) conjugated with horseradish peroxidase (Cat. Nos. 458 or 330; IB, 1/5000) from MBL; and anti-rabbit or mouse IgG (H+L) conjugated with Alexa Fluor 488 (Cat. Nos. A-11008 or A-11001; IF, 1/500) and anti-rabbit or mouse IgG (H+L) conjugated with Alexa Fluor 594 (Cat. Nos. A-11012 or A-11005; IF, 1/500) from Thermo Fisher Scientific (Waltham, MA, USA).

### 2.2. Plasmid Constructions

The full-length mouse *eprs1* (GenBank Acc. No. NM_029735) gene inserted into a pCMV6-based vector (FLAG- and myc-tagged *eprs1* expression plasmid, Cat. No. MR218237) was purchased from Origene (Rockville, MD, USA).

The *eprs1* plasmid harboring the Arg339-to-Ter (R339X [1015C-to-T at the nucleotide level]; OMIM ID 617951) mutation was generated by Fujifilm (Tokyo, Japan). Its DNA sequence was confirmed by the Fasmac sequencing service (Kanagawa, Japan).

### 2.3. Cell Culture and Differentiation

African green monkey kidney epithelial cell-like COS-7 cells (Human Health Science Research Resource Bank, Osaka, Japan) were cultured on cell culture dishes (Greiner, Oberösterreich, Germany) in a culture medium consisting of Dulbecco’s Modified Eagle Medium (DMEM, Thermo Fisher Scientific) containing 10% heat-inactivated FBS and PenStrep reagent (Thermo Fisher Scientific) in 5% CO_2_ at 37 °C [12,13,14,15].

Cells from the oligodendroglial FBD-102b cell line (a mouse brain neuronal stem cell line) were kindly provided by Dr. Y. Tomo-oka (Tokyo University of Science, Chiba, Japan). These FBD-102b cells were cultured on cell culture dishes in a culture medium consisting of DMEM/Nutrient Mixture F-12 containing 10% heat-inactivated FBS and PenStrep reagent in 5% CO_2_ at 37 °C [13,14,15]. To induce differentiation, FBD-102b cells were cultured for several days in the same culture medium without FBS on cell culture dishes (Greiner) with advanced TC polymer modification in 5% CO_2_ at 37 °C [13,14,15]. Cells with myelin-like wide membranes bearing multiple processes from the cell bodies, i.e., with cellular surface areas of 50-square-micrometers, were identified as differentiated [13,14,15].

### 2.4. Transfection

Cells were transfected with their respective plasmids using a ScreenFect A or ScreenFect A Plus transfection kit (Fujifilm) according to the manufacturer’s instructions. The medium was replaced 4 h after transfection. Transfected cells were generally used for experiments 48 h after transfection. The transfection efficiencies of COS-7 and FBD-102b cells were approximately 75% and 25%, respectively [15]. Since COS-7 cells had a higher transfection efficiency, they were used for the subsequent biochemical experiments.

We confirmed that COS-7 and FBD-102b cells were viable under each experimental condition by verifying that attached trypan-blue-incorporating cells made up less than 5% of all cells in each culture [15].

### 2.5. Confocal Microscopic Mages

Coverslips loaded with cells fixed with 4% paraformaldehyde or 100% cold methanol were blocked with Blocking One reagent (Nacalai Tesque, Kyoto, Japan). These were then incubated with primary antibodies followed by secondary antibodies conjugated with Alexa Fluor dyes. The coverslips on each slide glass was mounted with Vectashield reagent (Vector Laboratories, Burlingame, CA, USA). 

TIFF images were collected through a microscope equipped with a laser-scanning Fluoview apparatus (FV1000D or FV1200, Olympus, Tokyo, Japan) and processed using Fluoview software (Olympus, Tokyo, Japan). The resulting color images were analyzed in Image J software (Bethesda, MD, USA). Each image in each figure is representative of three independent experimental results.

### 2.6. Polyacrylamide Gel Electrophoresis and Immunoblotting

Cells were lysed in lysis buffer A (50 mM HEPES-NaOH, pH 7.5, 150 mM NaCl, 20 mM MgCl_2_, 1 mM phenylmethane sulfonylfluoride, 1 μg/mL leupeptin, 1 mM EDTA, 1 mM Na_3_VO_4_, 10 mM NaF, and 0.5% NP-40) [15,16]. For non-denatured and denatured conditions, the supernatants were incubated with non-denaturing sample buffer (also called native sample buffer; Nacalai Tesque, Kyoto, Japan) and denaturing sample buffer (Nacalai Tesque, Kyoto, Japan), respectively, after centrifugation [13,14,15]. The samples were then separated on non-denatured or denatured polyacrylamide gels (also called pre-made PAGE gels; Nacalai Tesque, Kyoto, Japan). The electrophoretically separated proteins were transferred onto polyvinylidene difluoride membranes (Merck-Millipore, Darmstadt, Germany) and blocked with Blocking One reagent, then immunoblotted with primary antibodies followed by secondary antibodies conjugated with HRP proteins. The bound antibodies were detected by means of X-ray film exposure using ImmunoStar Zeta reagent (Fujifilm)

Images were captured as TIFF files using a Canon LiDE 400 scanner (Canon, Tokyo, Japan) and processed using the accompanying driver software (Canon, Kyoto, Japan). The band pixels were measured in Image J software. Each image in each figure is representative of three independent experimental results.

### 2.7. Affinity-Precipitation Assay for Rab7 Regulatory Molecules

We next performed an affinity-precipitation assay with guanine-nucleotide exchange factor (GEF) CCZ1 as the Rab7-specific activator and GTPase-activating protein (GAP) TBC1D5 as the Rab7-specific inactivator, using lysis buffer A and homogenized cell lysates. To detect CCZ1 and TBC1D1, we gently mixed the supernatants with guanine-nucleotide-free or guanosine-triphosphate-binding Rab7 protein-absorbed protein G resin (Nacalai Tesque, Kyoto, Japan), respectively, after centrifugation [16,17,18,19]. Active GEFs preferentially bind to guanine-nucleotide-free GTPases involving Rab7, whereas active GAPs bind to guanosine-triphosphate-binding GTPases [16,17,18,19,20]. The affinity-precipitates were denatured, subjected to polyacrylamide gel electrophoresis, and blotted onto membranes for immunoblotting to detect CCZ1 and TBC1D1.

### 2.8. Statistical Analysis

Values are means ± SD from separate experiments. Intergroup comparisons were made according to unpaired Student’s *t*-test using Excel (Microsoft, Redmond, WA, USA).

A one-way analysis of variance (ANOVA) was followed by a Fisher’s protected least significant difference (PLSD) test as a post hoc comparison using StatPlus (AnalystSoft, Walnut, CA, USA). Differences were considered significant when *p* < 0.05. 

### 2.9. Ethics Statement

In vitro and in vivo gene recombination techniques were performed in accordance with a protocol approved by both the Tokyo University of Pharmacy and Life Sciences Gene and Animal Care Committee (Approved Nos. L18-04, L18-05, L19-04, L19-05, L20-04, and L20-04).

## 3. Results

### 3.1. The R339X Mutant Proteins of EPRS1 Are Localized in Rab7-Positive Vesicle Structures

To investigate whether the R339X mutant proteins of EPRS1 form protein aggregates in cells, we transfected the plasmid encoding EPRS1 harboring the R339X mutation or the wild-type into FBD-102b cells. Whereas the wild-type EPRS1 proteins were localized throughout the cell bodies (Figure 1A), approximately 80% of all R339X mutant proteins were gathered into punctate structures within the cells (Figure 1B,C). To precisely identify the organelle that corresponded to these structures, we stained cells expressing EPRS1 proteins harboring the R339X mutation with the respective antibodies against the endoplasmic reticulum (ER), Golgi body, and lysosome antigens. The ER antigen KDEL and the Golgi matrix protein 130 (GM130) antigen were not co-localized with the R339X mutant proteins (Figure 2A,B and Figure 3A,B). The lysosomal-associated membrane protein 1 (LAMP1) antigen, on the other hand, was co-localized with the R339X mutant proteins (Figure 4A–C). As a comparative experiment, we also stained cells expressing the wild-type EPRS1 proteins with the respective organelle marker antibodies. The wild-type proteins were not stained with any antibodies detecting the ER, Golgi body, or lysosome (see Appendix A), indicating that, while the R339X mutant proteins were localized with the LAMP1 antigen, the wild-type proteins were not.

LAMP1 antigens are also known to be present in Rab7- and Rab9-positive vesicles, which are late endosome markers [21]. Accordingly, we next explored whether the mutant proteins are also co-localized with Rab7- and/or Rab9-positive vesicles. We found that the proteins were primarily co-localized with Rab7 and only partially co-localized with Rab9 (Figure 5A–C and Figure 6A–C). Comparative experiments confirmed that the wild-type proteins were not stained with any antibody that detects Rab7 or Rab9 (see Appendix A).

### 3.2. The R339X Mutant Proteins of EPRS1 Form Protein Aggregates and Modulate Signals Regulating Rab7

We next asked whether the R339X mutant proteins formed protein aggregates as other HLD-associated mutant proteins do. We subjected COS-7 cell lysates transfected with plasmids encoding R339X mutant or wild-type EPRS1 to non-denatured or denatured polyacrylamide gel electrophoresis. The R339X mutant proteins in the non-denatured polyacrylamide gel were present as high-molecular-weight aggregates; the wild-type proteins, in contrast, corresponded to the molecular weight detected in the denatured polyacrylamide gel (Figure 7A,B). This suggested that the R339X mutant EPRS1 proteins formed polymeric aggregates while wild-type EPRS1 proteins did not.

Because we had observed that the R339X mutant proteins were localized in Rab7-positive vesicles, we examined whether expression of the mutant proteins in cells affected the signals regulating Rab7. Rab7 is known to be regulated primarily by the Rab7-specific GEF (activator) CCZ1 [22,23] and by the Rab7-specific GAP (inactivator) TBC1D5 [22,23]. Using affinity-precipitation assays, in which active GEFs preferentially bind to guanine-nucleotide-free GTPases whereas active GAPs bind to guanosine-triphosphate-binding GTPases [16,17,18,19], we found CCZ1 and TBC1D5 activities in FBD-102b cell lysates transfected with EPRS1 harboring the R339X mutation as well as those transfected with the wild-type. Expression of the mutant proteins in FBD-102b cells increased affinity-precipitated CCZ1 proteins (Figure 8A), but decreased affinity-precipitated TBC1D1 proteins (Figure 8B), indicating the upregulation of Rab7 signaling.

### 3.3. Expression of the R339X Mutant Proteins of EPRS1 in Cells Inhibits Morphological Differentiation

We explored the effects of the R339X mutant proteins of EPRS1 expression on oligodendroglial cell morphological differentiation. We expressed the R339X mutant or wild-type proteins in FBD-102b cells and allowed the cells to differentiate for 5 days. Expression of the mutant proteins in cells resulted in inhibited morphological differentiation: the proportion of myelin web-like structures bearing multiple processes was reduced by approximately 60% compared to the proportion in cells expressing wild-type proteins (Figure 9A,B). The inhibitory effects were supported by a decrease in the amount of myelin basic protein (MBP), the major myelin marker protein (Figure 9C). On the other hand, the oligodendrocyte lineage cell marker SOX10 and the internal control protein actin were comparable in the mutant- and wild-type-transfected cells.

## 4. Discussion

EPRS1 is a bifunctional aminoacyl-tRNA synthetase catalyzing the aminoacylation of glutamic acid and proline tRNA species. Human EPRS1 is a 172-kDa monomeric protein that is also known to be the substrate of protein kinases [24]. EPRS1 and its regulation by phosphorylation are involved in the formation of the γ-interferon-activated inhibitor of translation (GAIT) complex that regulates the translation of multiple genes [24]; EPRS1 is thus an essential molecule in the maintenance of basic cellular functions such as cellular differentiation as well as growth and homeostasis. Therefore, if a mutation in EPRS1 causes its dysfunction, it is conceivable that this mutation would also lead to serious disease. In keeping with this expectation, brain imaging in HLD15 patients with *eprs1* mutation reveals severe hypomyelinating leukodystrophy with thin corpus callosum [11]. Patients also exhibit symptoms such as dystonia, ataxia, spasticity, significant visual loss with variable amblyopia, and dysphagia. Because symptomatic treatment is typically the only treatment option available for HLD15 and other HLDs [11], it is necessary to investigate how HLD15 causes hypomyelinating leukodystrophy and its other symptoms at the molecular and cellular levels as knowledge of these pathological mechanisms may enable us to develop therapeutic treatment options and identify therapeutic target molecules.

Here we demonstrate that HLD15-assoociated R339X mutant proteins are localized as protein aggregates in Rab7-positive vesicle structures in FBD-102b cells. In contrast, wild-type proteins are present throughout cell bodies. These results are supported by our findings that the R339X mutant proteins, but not the wild-type ones, were co-stained with an anti-Rab7 antibody, a marker of late endosome, and that the mutant proteins had high molecular weight in non-denaturing polyacrylamide gel electrophoresis. Expression of the mutants, but not the wild-types, in cells stimulated signaling that regulates Rab7. In addition, cells expressing the wild-type proteins exhibited a phenotype consisting of myelin web-like structures bearing processes following the induction of differentiation, whereas cells expressing the R339X mutant proteins failed to exhibit this differentiated phenotype. These results were also supported by changes in the expression levels of myelin marker proteins. To summarize, HLD15-associated R339X mutation of EPRS1 proteins leads to protein aggregation in Rab7-positive vesicle structures and triggers signaling that regulates Rab7, inhibiting oligodendroglial cell morphological differentiation.

Human EPRS1 contains the D^926^XXD^929^ motif, which is cleaved by a caspase enzyme [25]. This motif is conserved among humans and other mammalian species such as mice and rats. After the protein has been cleaved, the N-terminal half has only the catalytic activity of glutamyl-tRNA synthetase (EARS), while the C-terminal half has only the catalytic activity of prolyl-tRNA synthetase (PARS). These caspase-cleaved EPRS1 half-proteins can also be incorporated into the functional tRNA multisynthetase complex (MSC), which is composed of aminoacyl-tRNA synthetases along with other adaptor proteins [25], suggesting that both the EARS unit and the PARS unit of EPRS1 are structurally and functionally required for MSC. Indeed, wild-type EPRS1 supports both EARS and PARS activities in cells. Since the respective HLD15-associated mutations of EPRS1 are point mutations in either the N-terminal half region or the C-terminal half region of EPRS1, they generally decrease either EARS or PARS enzymatic activity; however, the R339X mutation, however, results in an immature EPRS1 protein product and does not allow either EARS or PARS enzymatic activity. The loss of both activities due to the R339X mutation has a significant impact on survival [11]. From this point of view, the R339X mutation could be loss-of-function. Importantly, EARS2 and PARS2 proteins, which are thought to be localized in the mitochondria, are expressed in human and mammalian cells. EARS2 and PARS2 may compensate for EPRS1, at least in part [26,27,28]. This may be why the R339X mutation generates postnatal effects but does not necessarily cause embryonic lethality.

The R339X mutation of EPRS1 leads to protein aggregation in Rab7-positive vesicle structures, and it has been predicted that, as in other proteinopathies involving HLD1 [1,2,3,4], this protein aggregation leads to decreases cell viability and in turn to inhibits oligodendroglial cell morphological differentiation. From this point of view, the R339X mutation could be a toxic gain-of-function mutation. Further studies will allow us to determine whether the Rab7-positive vesicle localization of the R339X mutant proteins and the Rab7 signaling upregulation are directly or indirectly related. Upregulation of Rab7 signaling is also observed in the peripheral neuropathy known as Charcot–Marie–Tooth disease 2B (CMT2B) [20]. It is likely that Rab7 signaling is upregulated in connection with the causes of several neuropathies.

TBC1D5 is known to be regulated by mechanistic target of rapamycin complex 1 (mTORC1), although it remains unclear how Rab7 GEF CCZ1 is stimulated by upstream signaling molecules [29]. It is possible that the R339X mutant proteins affect the activities of mTORC1 to downregulate TBC1D5. It is worth noting that some HLDs other than HLD15 are associated with signaling through mTORC1 [13,15]. Molecular units centered on mTORC1 may constitute one of the common pathological mechanisms underlying HLDs involving HLD15.

Therapeutic treatments as opposed to than symptomatic treatments have been attempted in disease model mice. Previous studies in HLD1 model mice and their tissue levels have shown that ketogenic diet, neuroprotective chemicals, and stem cell transplantation are effective or potentially or effective therapeutic methods that may ameliorate hypomyelination and/or demyelination (see Table 2; Refs. [9,11]). Since these therapeutic methods are effective for HLD1 disease phenotypes but are not specific for their phenotypes, studies on therapeutic methods specific to therapeutic-target-molecule(s) are needed. Here, we report that an upregulation of Rab7 signaling is caused by HLD15-associated mutant proteins in cells. It is possible that modulation of the activities of Rab7 and Rab7 regulatory units, probably including mTORC1, by a chemical inhibitor or a therapeutic RNA harboring an siRNA backbone could provide a therapeutic method that works specifically against HLD15. Studies clarifying the molecular pathological mechanisms that are common to multiple HLDs as well as the mechanisms that are specifically involved in HLD15 will shed light on therapeutic methods that can be used against these rare diseases.

We previously reported that vascular cell adhesion molecule 1 (VCAM1), acting together with cluster of differentiation 69 (CD69), regulates oligodendrocyte myelination [30]. Interestingly, Bioinformatics shows that EPRS1 binds to VCAM1 (see BioGRID website, ULR: https://thebiogrid.org/, accessed on 10 January 2021). Changes in EPRS1’s binding to VCAM1 may be involved in one of the molecular and cellular pathological mechanisms that underlie HLD15. VCAM1 may directly and functionally contribute to the production of a number of proteins containing glutamic acid and proline functionally by interacting with EPRS1 or by determining EPRS1 localization in oligodendrocytes during myelination.
polymers-13-01074-t002_Table 2Table 2Treatments administered to HLD1 model mice in previous studies and their effects. Treatments, effects, and reference numbers are shown.DiseaseTreatmentEffectReference(See References)HLD1LonaprisanImprove the poor motor phenotype and increase in the number of myelinated axons[31]HLD1CurcuminImprove the poor motor phenotype and increase in the number of myelinated axons[32]HLD1Ketogenic dietRestore oligodendrocyte integrity and increase CNS myelination[33]HLD1Transplantation with neural stem cellsInduce neural regeneration in the CNS[34]HLD1DeferiproneInduce reduced oligodendrocyte apoptosis and enabled myelin formation[35]

## 5. Conclusions

We show here for the first time that the R339X mutant EPRS1 proteins, but not the wild-type EPRS1 proteins, form high-molecular-weight aggregates. The R339X mutant proteins are localized in Rab7-positive vesicle structures where they stimulate Rab7 regulatory signaling, inhibiting oligodendroglial cell morphological changes. Further studies will allow us to understand not only the detailed mechanisms by which the R339X mutant proteins stimulate Rab7 regulatory signaling but also the process by which expression of the R339X mutant proteins inhibits cell morphological differentiation. Together, such studies may reveal a common molecular mechanism underlying other HLDs as well as HLD15 and may enable the development of drug-target-specific medicines.

## Figures and Tables

**Figure 1 polymers-13-01074-f001:**
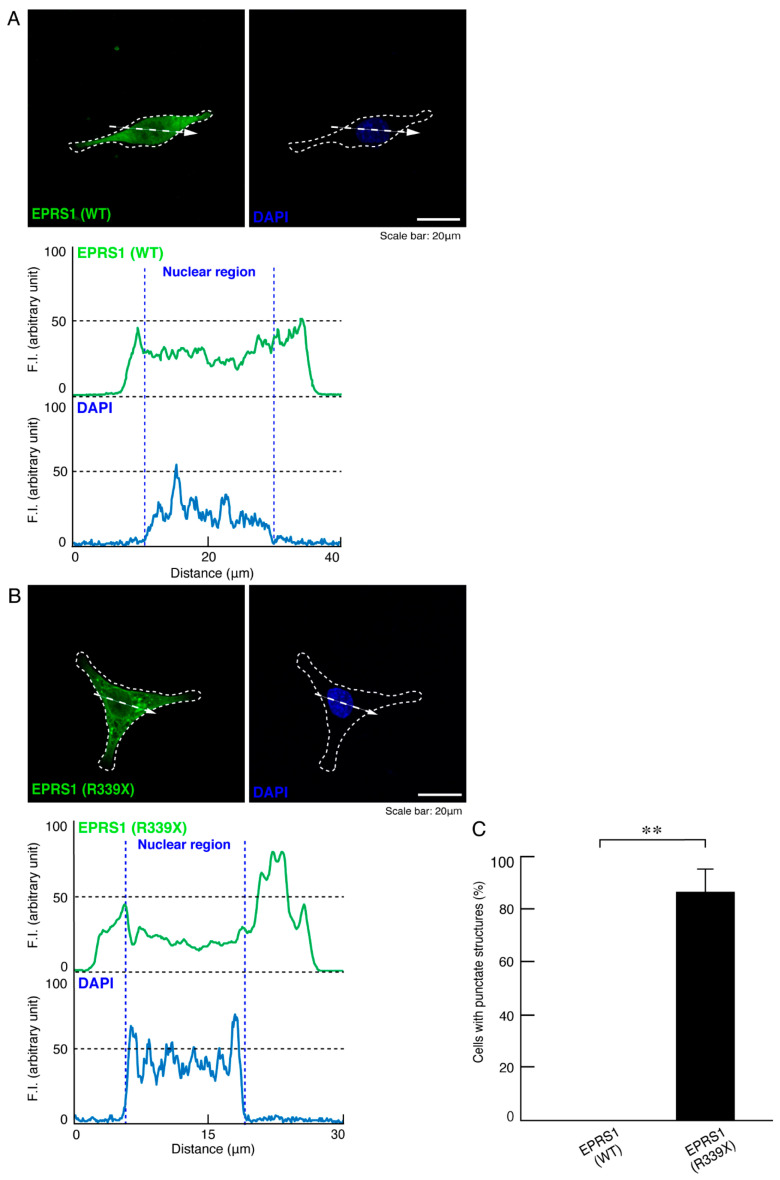
The R339X mutant EPRS1 proteins accumulate into punctate structures in cells. (**A**,**B**) FBD-102b cells, here surrounded by dotted lines, were transfected with the plasmid encoding the wild-type (WT) or the R339X mutant construct of FLAG-tagged EPRS1. Transfected cells were stained with an anti-FLAG antibody (green) and DAPI (blue). A scan plot was performed along the white line in the direction of the arrow in the image. Graphs showing the fluorescence intensities (arbitrary units) along the white lines can be seen in the bottom panels. (**C**) Percentages of cells with punctate structures were statistically assessed (** *p* < 0.01; *n* = 3 fields).

**Figure 2 polymers-13-01074-f002:**
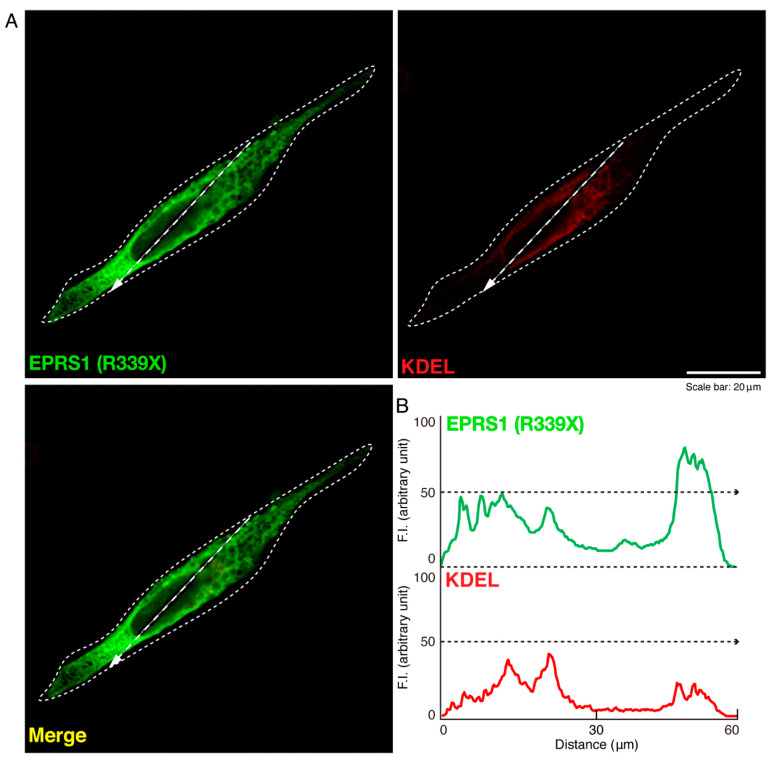
The R339X mutant EPRS1 proteins are not co-localized with the endoplasmic reticulum (ER) marker (KDEL). (**A**,**B**) FBD-102b cells were transfected with the plasmid encoding the R339X mutant construct of FLAG-tagged EPRS1 and stained with antibodies against the FLAG antigen (green) and the ER marker KDEL antigen (red). Scan plots were performed along the white dotted lines in the direction of the arrows in the color images (green and red). Graphs showing the fluorescence intensities (arbitrary units) along the white lines in the direction of the arrows can be seen in the bottom panels.

**Figure 3 polymers-13-01074-f003:**
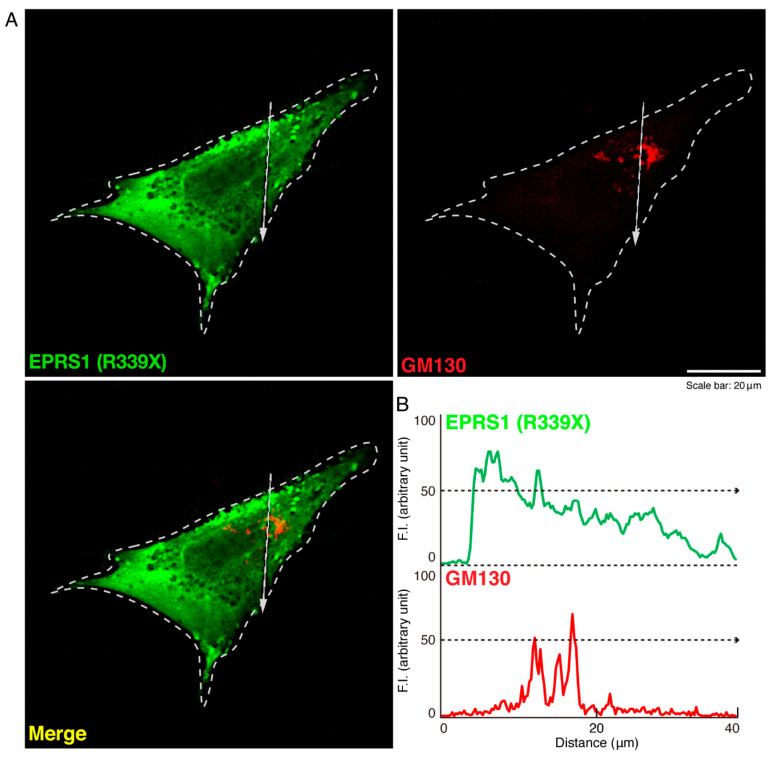
The R339X mutant EPRS1 proteins are not co-localized with the Golgi body marker (GM130). (**A**,**B**) FBD-102b cells were transfected with the plasmid encoding the R339X mutant construct of FLAG-tagged EPRS1 and stained with antibodies against the FLAG antigen (green) and the Golgi body marker GM130 antigen (red). Scan plots were performed along the white dotted lines in the direction of the arrows in the color images (green and red). Graphs showing the fluorescence intensities (arbitrary units) along the white lines in the direction of the arrows can be seen in the bottom panels.

**Figure 4 polymers-13-01074-f004:**
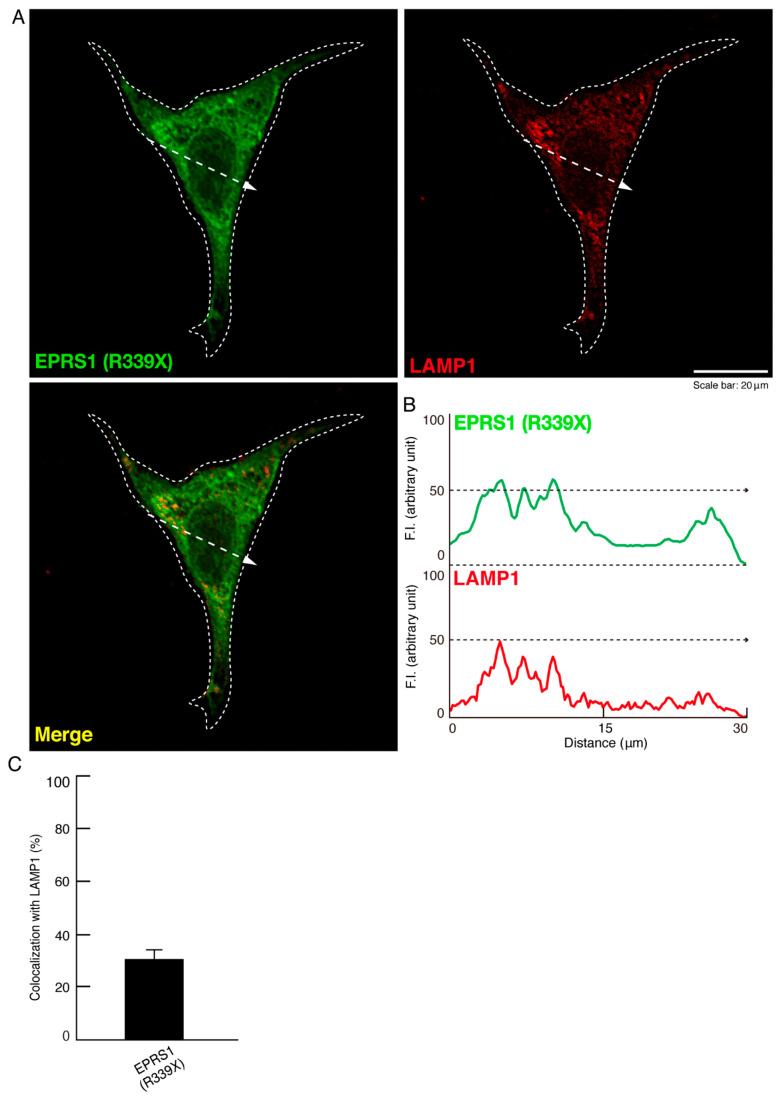
The R339X mutant EPRS1 proteins are co-localized with the lysosome marker (LAMP1). (**A**,**B**) FBD-102b cells were transfected with the plasmid encoding the R339X mutant construct of FLAG-tagged EPRS1 and stained with antibodies against the FLAG antigen (green) and the lysosome marker LAMP1 antigen (red). Scan plots were performed along the white dotted lines in the direction of the arrows in the color images (green and red). Graphs showing the fluorescence intensities (arbitrary units) along the white lines in the direction of the arrows can be seen in the bottom panels. (**C**) Merged percentages of mutant proteins with organelles are shown in the graph (*n* = 3 fields).

**Figure 5 polymers-13-01074-f005:**
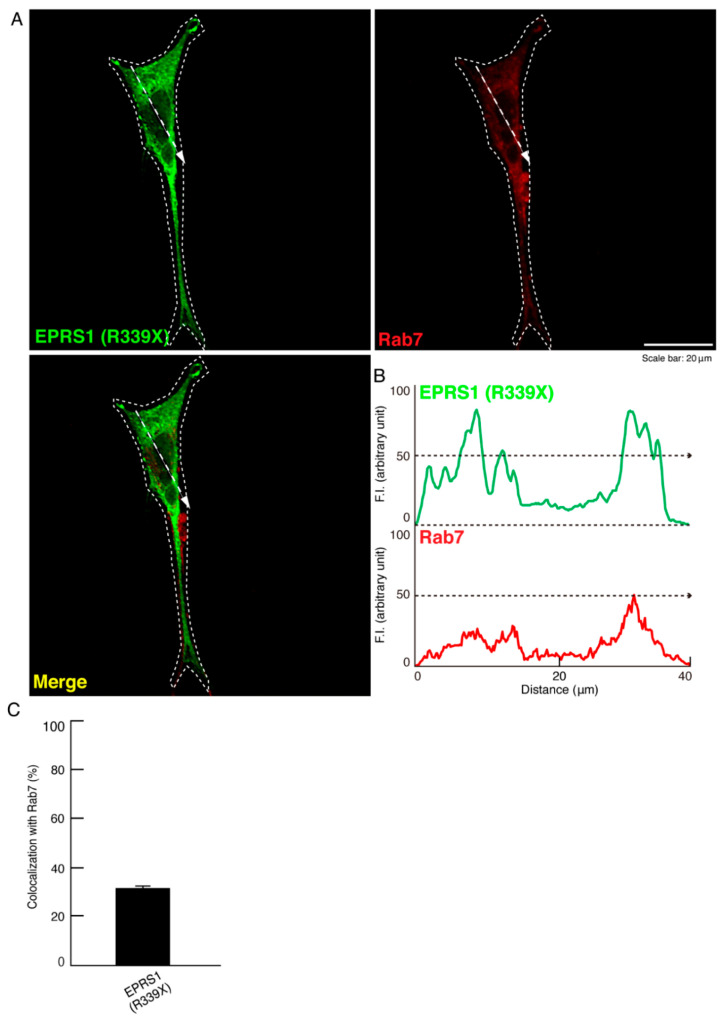
The R339X mutant EPRS1 proteins are co-localized with the early endosome marker (Rab7). (**A**,**B**) FBD-102b cells were transfected with the plasmid encoding the R339X mutant construct of FLAG-tagged EPRS1 and stained with antibodies against the FLAG antigen (green) and the early endosome marker Rab7 antigen (red). Scan plots were performed along the white dotted lines in the direction of the arrows in the color images (green and red). Graphs showing the fluorescence intensities (arbitrary units) along the white lines in the direction of the arrows can be seen in the bottom panels. (**C**) Merged percentages of mutant proteins with organelles are shown in the graph (*n* = 3 fields).

**Figure 6 polymers-13-01074-f006:**
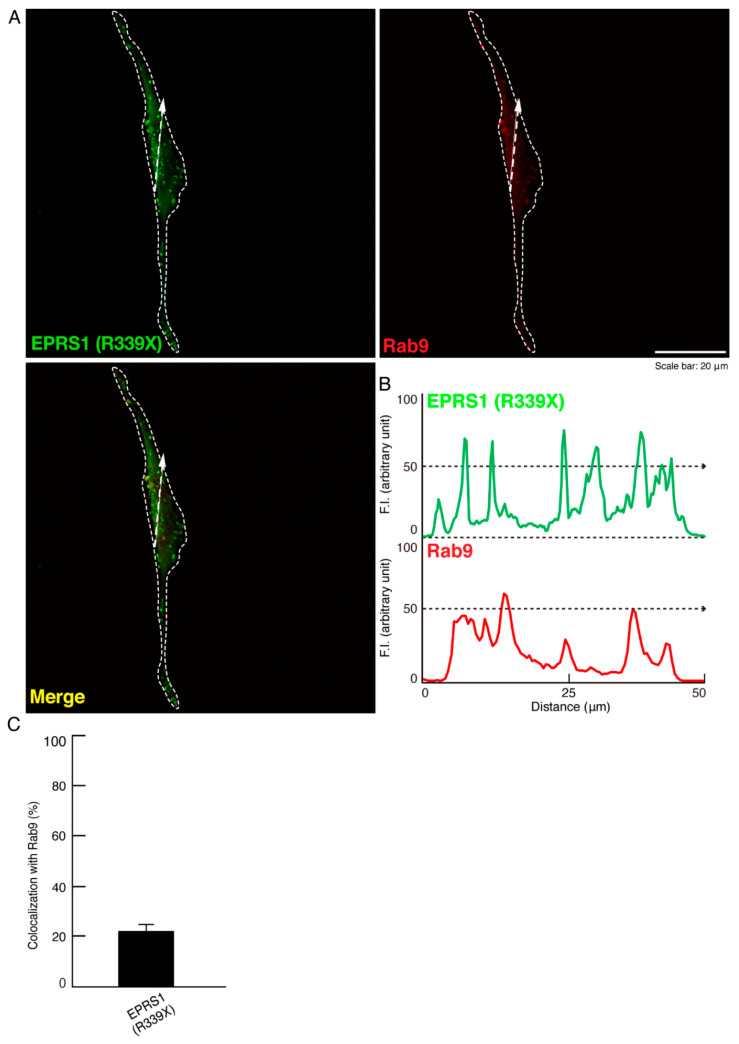
The R339X mutant EPRS1 proteins are partially co-localized with another early endosome marker (Rab9). (**A**,**B**) FBD-102b cells were transfected with the plasmid encoding the R339X mutant construct of FLAG-tagged EPRS1 and stained with antibodies against the FLAG antigen (green) and Rab9 antigen, another early endosome marker (red). Scan plots were performed along the white dotted lines in the direction of the arrows in the color images (green and red). Graphs showing the fluorescence intensities (arbitrary units) along the white lines in the direction of the arrows can be seen in the bottom panels. (**C**) Merged percentages of mutant proteins with organelles are shown in the graph (*n* = 3 fields).

**Figure 7 polymers-13-01074-f007:**
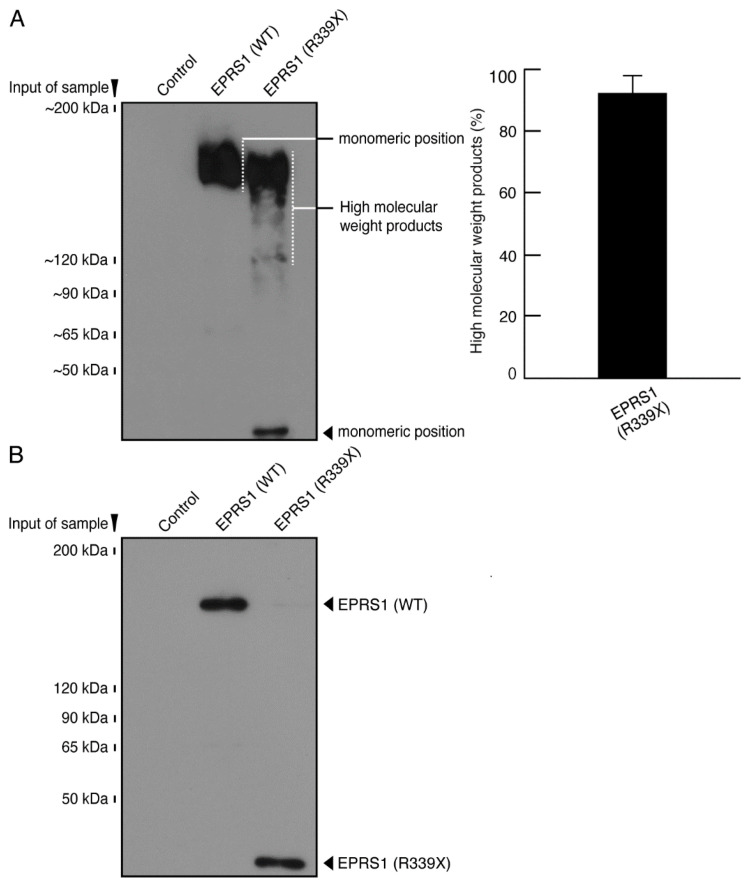
The R339X mutant EPRS1 proteins exhibit polymeric structures in non-denaturing polyacrylamide gel electrophoresis. (**A**) The lysates of COS-7 cells transfected with an empty vector or with a plasmid encoding the wild-type (WT) or R339X EPRS1 tagged with FLAG epitope were subjected to non-denaturing polyacrylamide gel electrophoresis and then immunoblotted with an anti-FLAG antibody. The position corresponding to the molecular weight of the wild-type EPRS1 monomer or the R339X EPRS1 monomer or polymer (including high-molecular-weight products) is shown. Abundance ratio (proportion) of high-molecular-weight products to the total proteins detected in the R339X lane is also shown (*n* = 3 blots). (**B**) As in the control experiments described in *A*, the lysates were also subjected to denaturing polyacrylamide gel electrophoresis and immunoblotted with an anti-FLAG antibody. The position corresponding to the wild-type or R339X EPRS1 monomer is shown.

**Figure 8 polymers-13-01074-f008:**
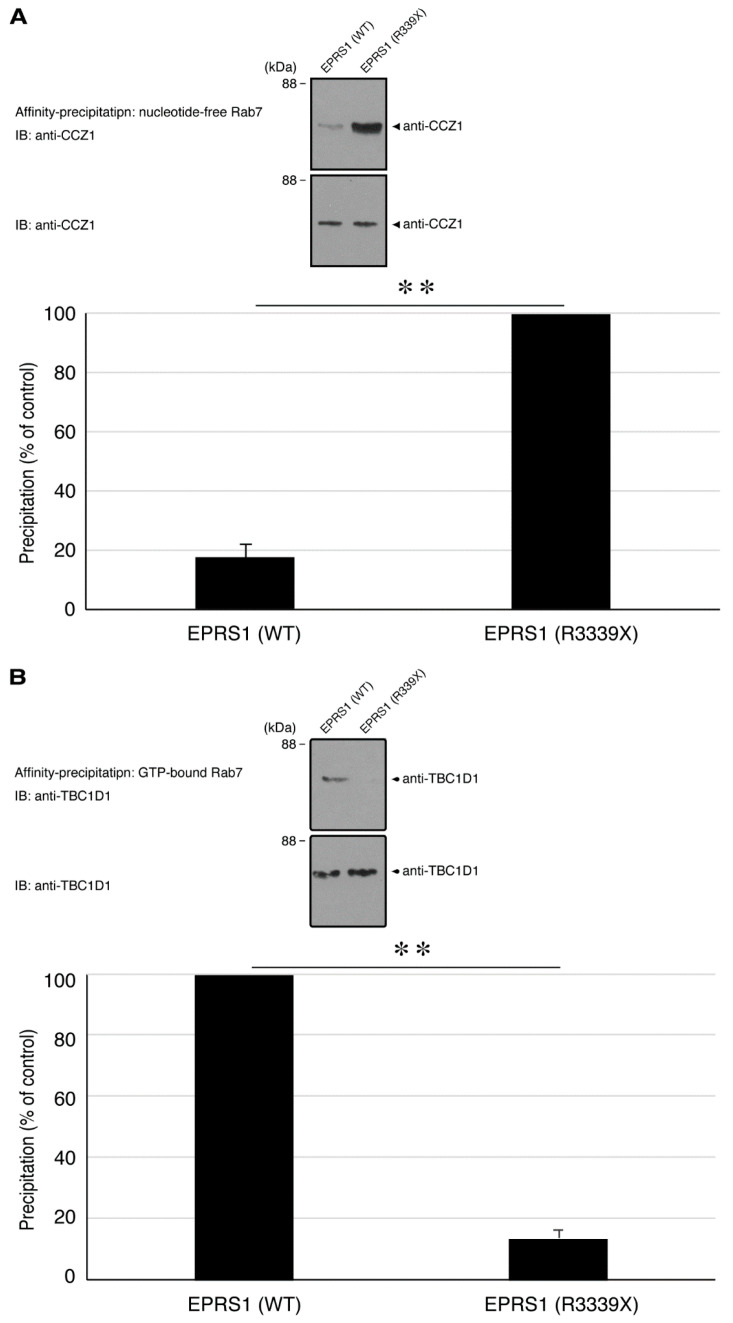
Expression of the R339X mutant EPRS1 proteins in cells increase CCZ1 and decreases TBC1D1. (**A**,**B**) FBD-102b cells were transfected with the plasmid encoding the wild-type (WT) or the R339X mutant construct of FLAG-tagged EPRS1. The lysates were affinity-precipitated using guanine-nucleotide-free or guanosine-triphosphate (GTP)-binding Rab7 protein for Rab7-GEF (activator) CCZ1 or Rab7-GTPase-activating protein (GAP) (inactivator) TBC1D1 and then immunoblotted with an anti-CCZ1 or TBC1D1 antibody, respectively. Affinity-precipitated CCZ1 or TBC1D1 proteins are compared as percentage and shown as percentages (** *p* < 0.01; *n* = 3 blots). Total CCZ1 and TBC1D1 proteins are also shown.

**Figure 9 polymers-13-01074-f009:**
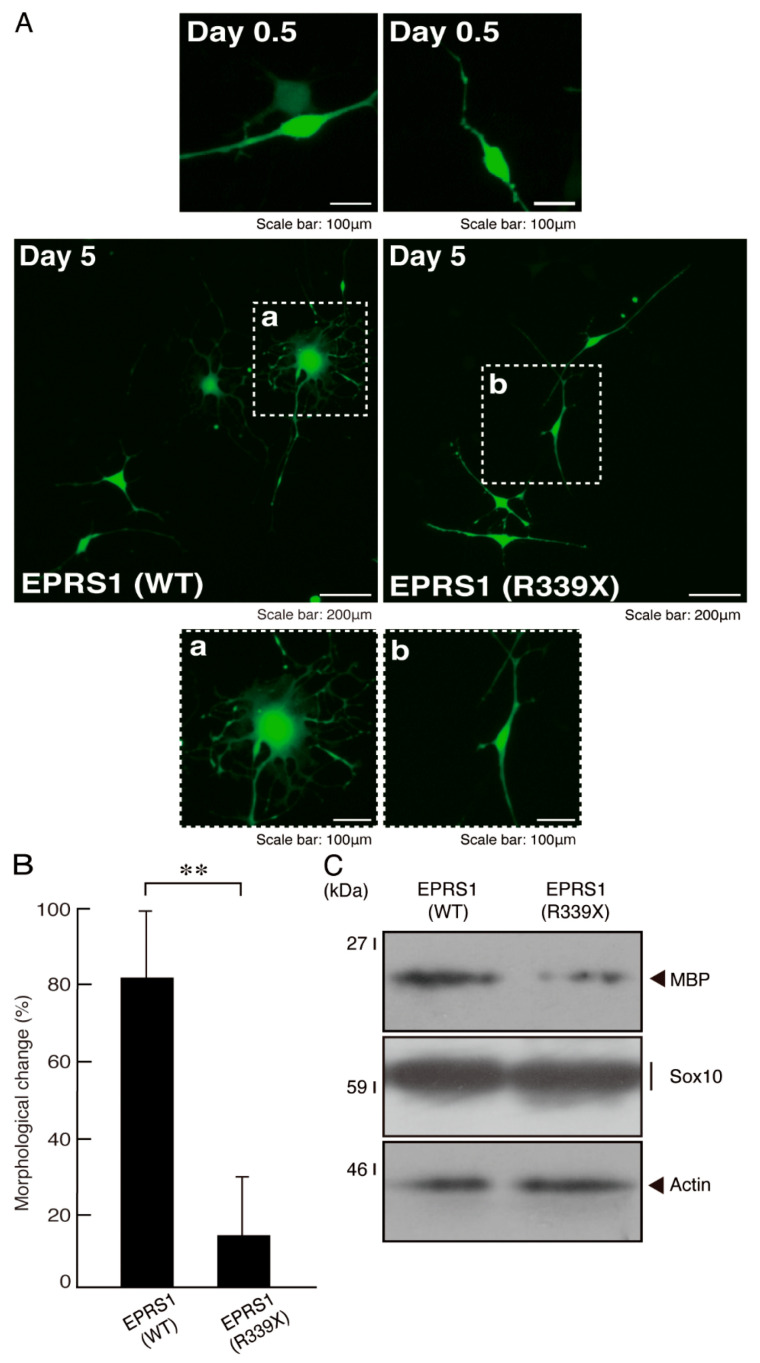
Cells expressing the R339X mutant EPRS1 proteins but not those expressing wild-type EPRS1 proteins fail to exhibit morphological differentiation. (**A**) FBD-102b cells expressing the wild type (WT) or the R339X mutant proteins of FLAG-tagged EPRS1 were allowed to differentiate for 5 days before fluorescence microscopic images were obtained (green color). Some cells are surrounded by white dotted lines. The square fields a and b indicated by dotted lines in the center panels are magnified in the bottom panels a and b. (**B**) Cells with myelin web-like membrane structures bearing multiple processes from the cell bodies were statistically assessed (** *p* < 0.01; *n* = 3 fields). (**C**) The lysates of the respective cells were immunoblotted with an antibody against MBP, Sox10 and control actin.

**Table 1 polymers-13-01074-t001:** The relationship between hypomyelinating leukodystrophies (HLDs) and their responsible genes. Types of HLDs, responsible genes, mutation forms, and numbers in the Online Mendelian Inheritance in Man (OMIM) catalog are shown.

Type	Responsible Gene	Mutation from	Number in OMIM
HLD1 (PMD)	*plp1*	Amplification, deletion, mutation, splice site mutation	312080
HLD2	*gjc2*	mutation, deletion	608804
HLD3	*aimp1*	mutation, deletion	260600
HLD4	*hspd1*	mutation	612233
HLD5	*fam126a*	mutation, splice site mutation	610532
HLD6	*tubb4a*	mutation	612438
HLD7	*polr3a*	mutation	607694
HLD8	*polr3b*	mutation	614381
HLD9	*rars1*	mutation	616140
HLD10	*pycr2*	mutation, splice site mutation	616420
HLD11	*polr1c*	mutation	616494
HLD12	*vps11*	mutation	616683
HLD13	*hikeshi*	mutation	616881
HLD14	*ufm1*	mutation, deletion	617899
HLD15	*eprs*	mutation	617951
HLD16	*tmem106b*	mutation, recurrent mutation	617964
HLD17	*aimp2*	mutation	618006
HLD18	*degs1*	mutation	618404
HLD19	*tmem63a*	mutation	618688
HLD20	*cnp*	mutation	619071

## Data Availability

The data presented in this study are available on request from the corresponding author.

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
