# Peer review of "Hypomyelinating Leukodystrophy 15 (HLD15)-Associated Mutation of EPRS1 Leads to Its Polymeric Aggregation in Rab7-Positive Vesicle Structures, Inhibiting Oligodendroglial Cell Morphological Differentiation"

_polymers, 2021, doi:10.3390/polym13071074_

Round 1
Reviewer 1 Report
Dear Authors,
I would like to congratulate on such well described manuscript and a research which was done very scrupulously. The manuscript is very clear and written properly, the methodology used in this manuscript is also very well described. I would like to include just three minor corrections for your consideration:
- Line 32 - there is an additional stop dot that should be removed.
- Line 43 - it would be very interesting to add a table with specific types of hypomyelinating leukodystrophies
- I would recommend checking the manuscript once again in terms of English since I have detected several mistakes and typos that should be corrected.
Best regards with your further research!
Reviewer
Author Response
Reviewer #1
We would like to thank the reviewer for referring to our paper as a “well described manuscript and a research which was done very scrupulously” and for stating that “The manuscript is very clear and written properly, the methodology used in this manuscript is also very well described...”.
- Line 32 - there is an additional stop dot that should be removed.
We sincerely thank the reviewer for pointing this out. We have removed the additional full stop.
- Line 43 - it would be very interesting to add a table with specific types of hypomyelinating leukodystrophies.
We agree with the reviewer’s comment and have added a new table (Table 1) outlining the relationship between the various types of HLDs and their responsible genes.
- I would recommend checking the manuscript once again in terms of English since I have detected several mistakes and typos that should be corrected.
We thank the reviewer for this comment. We have carefully corrected the typographical and grammatical errors throughout the manuscript and have had the entire paper checked by a native English speaker who specializes in editing scientific manuscripts.
Other modifications
- All sections in which significant portions of the text, as opposed to single words, have been changed are shown in red.
- The sentences in the Introduction, Results, and Discussion sections that were too similar to sentences in our previously published articles have been rewritten.
We are grateful for this reviewer’s input, which has enabled us to strengthen our argument for the proposed pathological roles of HLD15-associated mutant EPRS1 proteins at the molecular and cellular levels.
Reviewer 2 Report
In this manuscript the authors deal about hypomyelination leukodystrophy 15 (HLD15)-associated mutation of EPRS1 leads to its polymeric aggregation in Rab7-positive vesicle structures, inhibiting cell morphological differentiation.
First of all, there is an inconvenient regarding the similarity percent of this article 43%, which means a high overlap with other published articles; therefore the authors must reduce this percent (see attachment).
On the other hand, the manuscript is well structured and the quality of the figures and tables are satisfactory, the reference list covers the relevant literature adequately. That being said, I have the following comments and suggestions:
- A graphical abstract /general flowchart of the study is missing.
- In Introduction Section short data about pathophysiology, histopathology, potential treatment, differential diagnosis and prognosis of this very rare diseases should be added.
- Do the authors not have relevant conclusions for this research of theirs? I suggest adding the section “Conclusion”.
- Although the clinic of rare diseases is an extremely narrow therapeutic field, the authors must add in the discussion section the following key points: the therapeutic and clinical importance of this study, its novelty compared to the plethora of studies already published on this topic, the limits of this research and the points strong.
- Do the authors consider that this study brings an important contribution to the therapy of this very, very rare disease? If yes, justify this.
Consider revising accordingly!

Author Response
Reviewer #2
We would like to thank the reviewer for commenting that “...the manuscript is well structured and the quality of the figures and tables are satisfactory, the reference list covers the relevant literature adequately...”.
First of all, there is an inconvenient regarding the similarity percent of this article 43%, which means a high overlap with other published articles; therefore the authors must reduce this percent.
We sincerely thank the reviewer for pointing this out. The reason why a high percentage of text was similar to other articles was that our Methods section was similar to those of our own recent C11ORF73/HILESHI papers (Medicines [2020 and 2021]). We have changed various phrases and sentences throughout the manuscript to avoid overlap.
- A graphical abstract /general flowchart of the study is missing.
We thank the reviewer for this suggestion. We have made a graphical abstract/general flowchart of the study and uploaded it to the website.
- In Introduction Section short data about pathophysiology, histopathology, potential treatment, differential diagnosis and prognosis of this very rare diseases should be added.
We agree with the reviewer’s comments. We have added these details about HLD15 to the Introduction section.
- Do the authors not have relevant conclusions for this research of theirs? I suggest adding the section “Conclusion”.
We thank the reviewer for this comment. We have added a Conclusion section as the last section of the manuscript.
- Although the clinic of rare diseases is an extremely narrow therapeutic field, the authors must add in the discussion section the following key points: the therapeutic and clinical importance of this study, its novelty compared to the plethora of studies already published on this topic, the limits of this research and the points strong.
We thank the reviewer for these comments with which we agree. We have added information on the therapeutic treatments administered to HLD1 model mice to the Discussion section and listed the complete details in the new Table 2, with a particular focus on comparing previously published results with our own results in this study.
- Do the authors consider that this study brings an important contribution to the therapy of this very, very rare disease? If yes, justify this.
The reviewer raises an important question. We have added more information about the relevance of our results and how they can be applied for therapeutic purposes to the Discussion section. We also now discuss Rab7 as a possible drug target.
Other modifications
- All sections in which significant portions of the text, as opposed to single words, have been changed are shown in red.
- The sentences in the Introduction, Results, and Discussion sections that were too similar to sentences in our previously published articles have been rewritten.
We are grateful for this reviewer’s input, which has enabled us to strengthen our argument for the proposed pathological roles of HLD15-associated mutant EPRS1 proteins at the molecular and cellular levels.
Round 2
Reviewer 2 Report
No answer given.